# The Morphology Dependent Interaction between Silver Nanoparticles and Bovine Serum Albumin

**DOI:** 10.3390/ma16175821

**Published:** 2023-08-25

**Authors:** Jingyi Zhang, Xianjun Fu, Changling Yan, Gongke Wang

**Affiliations:** 1Henan Engineering Research Center of Design and Recycle for Advanced Electrochemical Energy Storage Materials, School of Materials Science and Engineering, Henan Normal University, Xinxiang 453007, China; 2Key Laboratory of Green Chemical Media and Reactions, Ministry of Education, Collaborative Innovation Center of Henan Province for Green Manufacturing of Fine Chemicals, School of Chemistry and Chemical Engineering, Henan Normal University, Xinxiang 453007, China

**Keywords:** silver nanoparticle, morphology, bovine serum albumin, protein corona

## Abstract

Biological applications of silver nanoparticles (AgNPs) depend on the covalently attached or adsorbed proteins. A series of biological effects of AgNPs within cells are determined by the size, shape, aspect ratio, surface charge, and modifiers. Herein, the morphology dependent interaction between AgNPs and protein was investigated. AgNPs with three different morphologies, such as silver nanospheres, silver nanorods, and silver nanotriangles, were employed to investigate the morphological effect on the interaction with a model protein: bovine serum albumin (BSA). The adsorptive interactions between BSA and the AgNPs were probed by UV-Vis spectroscopy, fluorescence spectroscopy, dynamic light scattering (DLS), Fourier transform infrared spectrometry (FTIR), transmission electron microscopy (TEM), and circular dichroism (CD) techniques. The results revealed that the particle size, shape, and dispersion of the three types of AgNPs markedly influence the interaction with BSA. Silver nanospheres and nanorods were capsulated by protein coronas, which led to slightly enlarged outer size. The silver nanotriangles evolved gradually into nanodisks in the presence of BSA. Fluorescence spectroscopy confirmed the static quenching the fluorescence emission of BSA by the three AgNPs. The FTIR and CD results suggested that the AgNPs with different morphologies had different effects on the secondary structure of BSA. The silver nanospheres and silver nanorods induced more pronounced structural changes than silver nanotriangles. These results suggest that the formation of a protein corona and the aggregation behaviors of AgNPs are markedly determined by their inherent morphologies.

## 1. Introduction

Silver nanoparticles (AgNPs) have attracted extensive attention due to their small-scale effects, antimicrobial activity, optical effects, biocompatibility, and excellent stability [1,2]. On basis of these characteristics, AgNPs are widely employed in the fields of biomedicine, environmental monitoring, biosensing, and chemical catalysis, etc. [3,4]. As AgNPs are among the most commonly used materials in biomedical research, it is important to study their interactions with proteins, which help more in-depth understanding of the biological effects and facilitate safe applications of AgNPs in biomedicine [5,6,7,8]. Upon incorporation into biological systems, AgNPs interact with proteins via various levels. Ultimately, it is the AgNPs–protein complexes that dictate the eventual biological responses rather than the AgNPs individually [9,10,11]. Therefore, the inspection into the interactions between AgNPs and proteins can offer essential guidance in design of nanomaterials for biological applications [12].

The tight and specific or non-specific bonding of proteins with nanomaterials constructs a surface coating layer designated as the protein corona [13,14], which underlies the biological identity on AgNPs and, thus, determines their functionality and reactivity [15,16,17]. The size, surface properties, morphology, chemical composition, surface hydrophobicity, and surface charge of AgNPs affect the adsorptivity and composition of the corona around AgNPs [18,19]. Up to now, the interaction of AgNPs with proteins were extensively studied. For example, Mariam et al. found that the interaction between bovine serum albumin (BSA) and AgNPs was spontaneous and mainly driven by hydrophobic forces, and the fluorescence of BSA could be quenched by AgNPs in dynamic as well as static quenching processes [20]. Sasidharan conducted systematic investigations on time-dependent adsorption kinetics and individual protein corona formation with citrate and lipoic acid-coated 40-nm gold nanoparticles (AuNPs) and AgNPs. The results showed that regardless of the composition and surface chemistry of the nanoparticles (NPs), HSA and IgG showed strong binding to both AuNPs and AgNPs [21]. In addition, Zhang and coworkers synthesized citrate-coated AgNPs with three different sizes to investigate the influence of size on their biological effects, and concluded that the interaction between AgNPs and HSA was related to the size of the AgNPs. Furthermore, smaller AgNPs showed stronger binding interaction to HSA relative to larger ones at the same concentration [22]. Sajid Ali’s group designed a novel polyvinylthiol coated silver nanoparticle (Ag-PVT) and evaluated the interaction of Ag-PVT with HSA, which revealed the formation of 1:1 ground state complex and the crucial role of hydrophobic forces in stabilizing the complex [23]. Despite these progresses, seldom studies involved the morphology dependence of AgNPs on the interactions with proteins, which is meaningful for further biomedical applications of AgNPs.

As the most abundant protein in bovine plasma, BSA is reversibly bonded to a large variety of compounds and is a significant carrier of fatty acids and metabolites, such as thyroid hormone, endogenous, bilirubin, exogenous compounds, and so on. BSA plays an important role in oxidative stress because its sulfhydryl groups act as scavengers of reactive oxygen species and nitrogen [24]. BSA is formed by 582 amino acid residues with a molecular weight of 69,000, two tryptophan residues at positions 134 and 212. Therefore, we chose BSA as model protein due to its major biological functions, which are important for interaction studies [25].

Herein, AgNPs with three different morphologies, i.e., silver nanospheres, silver nanorods, and silver nanotriangles, were synthesized to investigate the morphology dependent interactions with BSA in aqueous solutions. The changes in protein structure and AgNPs morphology upon adsorption, and the resulting stability of NP-protein conjugates, were characterized using a number of methods. Comprehensive understanding of AgNPs–protein interactions will facilitate further development of functionalized and safe biomedical applications of AgNPs.

## 2. Materials and Methods

### 2.1. Materials

Trisodium citrate (Na_3_C_6_H_5_O_7_, ≥99.0%), sodium borohydride (NaBH_4_, ≥99.99%), and BSA were purchased from Alfa Aesar (Ward Hill, MA, USA). Cetyltrimethylammonium bromide (CTAB, ≥99.0%), silver nitrate (AgNO_3_, ≥98.0%), l-ascorbic acid (AA, ≥99.7%), hydrogen peroxide (H_2_O_2_, 30 wt%), and sodium hydroxide (NaOH, ≥96.0%) were obtained from Shanghai Sinopharm (Shanghai, China). Polyvinylpyrrolidone (PVP, Mw ≈ 29,000) was obtained from Sigma-Aldrich (St. Louis, MO, USA). All chemicals were used as received without further treatment. All glassware was cleaned with aqua regia prior to use, and deionized water was used in all experiments. All procedures were carried out in phosphate-buffered saline (PBS, 0.01 M, pH 7.4).

### 2.2. Synthesis of AgNPs with Different Morphologies

Citrate-stabilized silver nanospheres were synthesized by citrate reduction [26]. Briefly, 18 mg of AgNO_3_ was dissolved in 100 mL of distilled water at 45 °C, stirred and heated rapidly to boiling such that the solution swirls to the bottom of the stirrer. Immediately upon the start of boiling, 2 mL of 1.0% sodium citrate solution was added under continuous stirring. The colorless solution changed to yellowish and finally to yellow-green within 15 min. After being held under boiling and stirring conditions for 15 min, the solution was cooled to room temperature under continuous stirring and sealed in a refrigerator at 4 °C for subsequent use.

Silver nanorods were prepared by the seed-mediated growth approach [27]. In a typical procedure, 0.5 mL of 0.01 M aqueous of AgNO_3_ and 0.01 M Na_3_C_6_H_5_O_7_ solution in 19 mL of H_2_O at 0 °C. Under vigorous stirring, 0.6 mL of 0.01 M NaBH_4_ aqueous solution containing 0.1 M NaOH solution was added immediately. After 1 min, the color of the solution turned into light yellow, indicating the formation of 4-nm seeds, which was kept at 25 °C for 2 h prior to use.

For the synthesis of silver nanorods, a reaction solution was prepared containing 1 mL of 10 mM AgNO_3_, 2 mL 100 mM ascorbic acid, and 20 mL 100 mM CTAB. Subsequently, 0.5 mL of 4 nm seed solution and 2 mL of 0.2 M NaOH solution were added. Gently shake the reaction vessel to mix the solution well and place it in a water bath at 25 °C for 10 min. The solution changed rapidly from achromatic to light yellow, brown, and brownish-red. The solution was centrifuged twice (8000 rpm, 30 min) to remove excess CTAB and spheres. The precipitate was dispersed in Milli-Q water and sealed in a refrigerator at 4 °C prior to use.

Silver nanotriangles were synthesized by a one-step method [28]. Typically, 21.74 mL of H_2_O, 50 μL of 0.05 M AgNO_3_, 0.5 mL of 75 mM Na_3_C_6_H_5_O_7_, 2.5 mL of 0.7 mM PVP, and 60 μL of H_2_O_2_ (30%) were mixed in a 100 mL beaker and stirred vigorously at room temperature in air. After injecting 250 μL of 100 mM NaBH_4_ solution for 2 min, the color changed to light yellow. The reaction was continued for 30 min and a color change from yellow to blue was observed, which indicated the formation of triangular silver NPs. In order to avoid the oxidation and aggregation of AgNPs, the stock solution of AgNPs was sealed in a refrigerator at 4 °C before used.

### 2.3. UV-Vis Measurements

UV-Vis were recorded on a spectrophotometer equipped with a xenon lamp and a quartz cuvette with a path length of 1.0 cm (TU-1810; Persee, Beijing, China). In this experiment, silver nanospheres (0.26 nM), silver nanorods (0.46 nM), and silver nanotriangles (0.12 mM) were measured in the spectral range of 300~900 nM in the presence of a constant concentration of BSA.

### 2.4. DLS Measurements

The hydrodynamic diameters of AgNPs with different morphologies were measured in the presence or absence of BSA using the Zetasizer Nano ZS (Malvern Instruments, Malvern, UK). To measure the size distribution, samples were diluted with Milli-Q water in cuvettes with a path length of 1.0 cm. The mixture was incubated for 30 min and dynamic light scattering (DLS) measurements were taken three times.

### 2.5. Fluorescence Spectroscopy Measurements

Fluorescence spectroscopy measurements were made using a spectrofluorometer (CARY Eclipse; Varian, Palo Alto, CA, USA) with a 1 cm × 1 cm quartz cuvette. The excitation wavelength of the protein and mixture solution (λex) was set at 280 nm, and the emission spectra of BSA (2 μM) were recorded in the presence or absence of AgNPs with different morphologies, at 298 K in the wavelength range of 300~550 nm. All measurements were repeated in triplicate.

### 2.6. CD Spectroscopy Measurements

Circular dichroism (CD) measurements were performed at room temperature on a Chirascan spectropolarimeter (Applied Photophysics Ltd., Leatherhead, UK) using a quartz cell with a path length of 1 cm. The CD spectra were recorded in the range of 190~260 nm with a bandwidth setting of 1 nm. PBS (0.01 M, pH 7.4) was used as a blank, alone without protein. The spectra of BSA solution with the concentration kept at 10 mM were measured in the absence and presence of AgNPs with different morphologies. The final spectra were calculated by deducting the buffer contribution from the original protein spectra and recorded as the average of three scans.

### 2.7. FTIR Spectroscopy Measurements

Fourier transform infrared spectrometry (FTIR) measurements were performed using an Avatar 360 ESP instrument (Thermo Scientific, Waltham, MA, USA). All spectra were obtained using the attenuated total reflection (ATR) method, with a resolution of 4 cm^−1^ and 60 scans. The FTIR spectra of BSA (10 mM) in the absence and presence of AgNPs in PBS (0.01 M, pH 7.4) were recorded in the range of 1900~1200 cm^−1^. The corresponding absorbance contributions of buffer and free AgNP solutions were recorded and digitally subtracted under the same conditions.

### 2.8. TEM Characterization

AgNPs of different morphologies were visualized in the absence and presence of BSA by transmission electron microscopy (TEM) (FEI Tecnai 12 BioTwin; FEI, Lausanne, Switzerland). Samples were diluted prior to TEM observation and a drop from each colloid was then placed onto a carbon-coated copper grid (CF-400 Cu; Electron Microscopy Sciences, Hatfield, PA, USA). Excess suspension was removed using filter paper.

## 3. Results and Discussion

### 3.1. Characterization of AgNPs

The different morphologies of AgNPs were characterized by UV-Vis absorption spectroscopy, DLS, and TEM, and are illustrated in Figure 1 and Figure 2. As shown in Figure 1, silver nanospheres have maximum absorbance at 420 nm. For silver nanorods, the absorption band with two strong absorption peaks: one is a typical transverse surface plasmon resonance (SPR) of rod-shaped AgNPs centered at ca. 425 nm; the other is a longitudinal SPR centered at ca. 520 nm appears in the longer wavelength region. The two strong absorption peaks appearing simultaneously in the longer and shorter wavelength regions were the characteristic peaks of metal nanorods [29,30]. The characteristic absorption demonstrated that the NPs prepared by this reaction system were mainly silver nanorods. Furthermore, three distinct peaks at 332, 483, and 728 nm were observed in the absorption spectra, which are characteristic of silver nanotriangles. According to previous reports, these three peaks can be attributed to the out of-plane quadrupole, in-plane quadrupole, and in-plane dipole plasmon resonance modes, respectively [31,32]. The appearance of these three peaks clearly indicated the formation of silver nanotriangles. TEM confirmed successful formation of silver nanospheres, silver nanorods, and silver nanotriangles (Figure 2A–C). The average dimensions of the AuNPs were determined by counting 80 NPs in TEM via NanoMeasurer software. The sizes of AgNPs with three morphologies were also measured by DLS. As shown in Figure 2D–F, the mean diameters of silver nanospheres, length of silver nanorods, and size of silver nanotriangles were 85.66, 87.28, and 71.55 nm, respectively.

### 3.2. Fluorescence Spectroscopy

The interaction of AgNPs with BSA was monitored by fluorescence spectroscopy. The application of fluorescence spectroscopy for the study on the structure and conformation of proteins has proven to be fruitful [20,22,25]. The fluorescence spectra of BSA in the presence of different concentrations of AgNPs with three morphologies are shown in Figure 3. The fluorescence emission intensity of BSA at around 350 nm decreases obviously with the concentration of silver nanospheres, silver nanorods, and silver nanotriangles. The prominent decrements in maximum emission peak of three morphology AgNPs illustrate that the relative fluorescence quenching is increasing progressively with protein concentration, since the amount of adsorbed protein approximately follows this trend. Stronger bonding of the proteins to the NPs may also give rise to conformational changes of protein, where more of the amino acids of the bonded proteins are in proximity with the surface, and result in more efficient fluorescence quenching. Consequently, the interaction of the AgNPs with BSA changes the secondary structure of protein, leading to the changes in the tryptophan environment of BSA, which may be due to the formation of a complex between AgNPs and BSA [33]. At the same time, it is found that the fluorescence peak wavelength of BSA at 350 nm upon addition of silver nanospheres had an obvious red shift; silver nanorods caused a blue shift of the peak of BSA at 350 nm, while silver nanotriangles did not change the peak position of BSA at 350 nm. The shift of the fluorescence peak wavelength of the fluorescence emission spectrum in Figure 3 also contains significant information about the protein layer adsorbed on AgNPs. The blue or red shift of this feature is symptomatic of a shift of the dielectric properties of the medium, or more specifically a reduction or increase of the polarity of the local environment of the emitter species. Evidently, the local dielectric environment within the fully developed adsorbed protein layers is less or more polar than the corresponding emitter environment of the protein dispersed in solution [23]. Thus, the three morphologies of AgNPs had different effects on the secondary structure of BSA. In addition, silver nanospheres and silver nanorods had a much greater influence on the structure of BSA than silver nanotriangles.

The quenching mechanisms between quencher and protein can be classified into static and dynamic mechanisms. Dynamic quenching is the process of contacting a fluorophore with a quencher in a transient excited state, making the bimolecular quenching constant larger at higher temperatures. While static quenching is the consequence of the formation of a base-state complexes between a fluorophore and a quencher, the stability and static quenching constants decreases with an increase in temperature. The main mechanism can be determined by comparing the value of the biomolecular quenching constant (*K_q_*) and the limiting diffusion rate constant of biomolecules (*K_d_* ≈ 2.0 × 10^10^ M^−1^ S^−1^) [33]. The magnitude of *K_q_* and *K_d_* determine the dominant mechanism of the quenching, when *K_q_* < *K_d_*, dynamic quenching is the dominant mechanism; otherwise, static quenching plays a major role. The Stern–Volmer equation is calculated to obtain the quenching Mode (1).
(1)$$\frac{F_{0}}{F}=1+k_{q}\tau_{0}\left[Q\right]=1+K_{SV}\left[Q\right]$$
where *F*_0_ (and *F*) denotes the fluorescence intensity of protein in the absence AgNPs (and in the presence of AgNPs). *K_SV_* is the Stern–Volmer quenching constant, *K_q_* is the biomolecular quenching constant, and *τ*_0_ is the average lifetime of the excited biomolecule without a quencher (generally 10^−8^ s) [34]. [*Q*] is the quencher concentration (AgNPs here). The values of *K_SV_* and *K_q_* for different BSA–AgNPs interaction systems, obtained from Figure 3, are listed in Appendix A. The values of *K_q_* in the different BSA–AgNPs interaction systems were all higher than the limiting diffusion rate constant of biomolecules (*K_d_*), indicating that static quenching should occur in the interaction of the three different morphological types of AgNPs with BSA. Static quenching suggested the formation of ground-state complexes between BSA and AgNPs with different morphologies.

Particle size, morphology, surface charge, and other thermodynamic factors that affect the strength of protein–NP binding should likewise influence the relative value of the protein–NP binding constant *K*, which quantifies the relative strength of the protein–NP binding. The synergistic and binding constants of the interaction between BSA and AgNPs with different morphologies can be determined by the Hill Equation (2):(2)$$\log\frac{F-F_{\min}}{F_{0}-F}=m\log K_{D}-mlog\left[Q\right]$$
where *F*_min_ is the minimum fluorescence intensity of protein in the absence of AgNPs, *K_D_* is the dissociation constant of the interaction between AgNPs and BSA, and *m* is the Hill coefficient. If *m* > 1, there is a positive synergistic effect, indicating that the interaction between protein and AgNPs can promote their further binding, while if *m* < 1, there is a negative synergistic effect, indicating that the interaction between protein and AgNPs will prevent subsequent protein adsorption onto the surface of AgNPs. If *m* = 1, there is no synergistic effect, so the adsorption of protein on the surface of AgNPs does not affect the subsequent adsorption of protein. *F*_0_, *F*, and [*Q*] have the same meanings as in Equation (1). *K_D_* and *m* can be obtained by calculating the intercept and slope of best-fitting line in a double-log plot of log[(*F* − *F*_min_)/(*F*_0_ − *F*)] vs. log[*Q*] (Figure 3). The calculated *m* and *K_D_* values are presented in Table 1.

Generally, the adsorption of protein onto the negatively charged NP must reduce the electrostatic binding energy and, thus, the relative magnitudes of the enthalpy and entropies of protein binding; therefore, rationalizing the anti-cooperativity of the protein adsorption on the NPs. If the NPs induce the proteins to organize at their boundaries, on the other hand, we may naturally expect an enhancement of the cooperativity of protein absorption on the NPs. As shown in Table 1, the Hill coefficients of the interaction between BSA and AgNPs with different morphologies were all well over 1, suggesting the positive synergistic effects in binding of BSA to different AgNPs, in turn indicating that protein adsorbed on the surface of AgNPs effectively promotes subsequent protein adsorption.

### 3.3. Conformational Changes of Protein

The biological functions of proteins are dependent on their molecular conformation. Therefore, CD experiments were carried out to probe the conformational changes of BSA molecules during the binding process. Figure 4 shows the CD spectra of BSA bonded to AgNPs with different morphologies in PBS (pH = 7.4) buffer solution. The CD spectrum of native BSA showed two negative bands at 208 and 222 nm, characteristic of α-helix structures [35]. The negative peaks at 208 and 222 nm both contribute to n–π* transition for peptide bonds with α-helicity. As shown in Figure 4, both of these bands were slightly increased in the presence of silver nanospheres and silver nanotriangles, indicating the increased fraction of α-helix structures of BSA. However, the two characteristic bands were markedly reduced with the addition of silver nanorods, indicating the decreased fraction of α-helix structures within BSA. The changes in structure of BSA could be estimated by monitoring the α-helix fraction of the protein. The α-helix content of BSA was calculated using the software supplied with the CD spectropolarimeter, and the results are summarized in Appendix A. Native BSA was shown to have a high α-helix content of 58.4%, which increased to 67.5% and 60.2% by incorporation of silver nanospheres and silver nanotriangles, respectively, while declined to 43.8% by silver nanorods. In comparison to silver nanotriangles, both silver nanospheres and silver nanorods induced greater changes in the protein secondary structure. These observations showed that AgNPs associate with amino acid residues on the surface of BSA [36]. However, CD spectra of BSA were similar in the presence and absence of AgNPs, demonstrating that the structure of bovine serum albumin is also dominated by the α-helical structure. Thus, these observations indicated that binding between BSA and AgNPs leads to different changes in secondary structure of BSA molecules but did not significantly alter the conformation of the protein.

The conformational changes of BSA by AgNPs were further verified by FTIR. Typically, the peaks appearing within 1600~1700 cm^−1^ and 1548 cm^−1^ represent the amide I band (mainly C=O stretch) and amide II band (C–N stretch coupled with N–H bending mode). Amide I band promotes the exposure of α-helix structure, while amide II band corresponds to the β-sheet of the protein [37]. FTIR of BSA in different concentrations and different AgNPs are shown in Figure 5. Native BSA shows an exposed α-helix peak at 1654.63 cm^−1^, and the peak at 1548.56 cm^−1^ corresponds to the β-sheet structure. Upon incorporation of silver nanospheres, the exposed α-helix peak shifts to 1652.70 cm^−1^, while the β-sheet peak at 1548.56 cm^−1^ maintains unvaried (Figure 5A). These observations suggest that the α-helix is slightly changed, while the β-sheet remains intact. As can be seen in Figure 5B,C, the peak corresponding to β-sheet shows similar shifts to 1546.63 cm^−1^ by incorporation of silver nanorods and silver nanotriangles, while the exposed α-helix peak at 1654.63 cm^−1^ shifted to 1650.77 cm^−1^ with the addition of silver nanorods, but to 1652.70 cm^−1^ in the presence of silver nanotriangles. These observations indicate that silver nanorods and silver nanotriangles exert different effects on the secondary structure of BSA. Based on these results, the changes of secondary structure of BSA are determined by the morphological types of AgNPs; the trends concide with the results of CD.

### 3.4. Morphological Analysis of Three BSA-AgNPs Systems

The surface ligands for spherical, rod-shaped, and triangular AgNPs are sodium citrate, CTAB, and PVP, respectively. To elucidate the effects of the different surface ligands of the AgNPs (spherical, rod-shaped, and triangular) on their interactions with BSA, fluorescence measurements were adopted. As shown in Appendix A, when adding only the three surface ligands in the protein solution, it is seen that the fluorescence of BSA was slightly quenched (curves 2–4). However, with the addition of spherical, rod-shaped, and triangular AgNPs, the fluorescence of BSA decreased greatly (curves 5–7). Therefore, these results demonstrate that the influence of surface ligands around AgNPs to the interaction with BSA is negligible.

TEM was performed to confirm the morphology of silver nanospheres, silver nanorods, and silver nanotriangles interacting with BSA. The TEM images clearly showed an obvious protein corona on the surface of silver nanospheres with the addition of higher concentrations of BSA (Figure 6A). BSA was gradually adsorbed on the sides and tips of the silver nanorods, and no obvious protein corona was formed around the nanorods (Figure 6B). In sharp contrast, the morphology of the silver nanotriangles changed obviously after interaction with BSA; the “tips” became smooth and the overall morphology evolved from nanotriangles to nanodisks in the presence of BSA (Figure 6C). This is because the silver atoms at the tips of the nanotriangles were extremely unstable. Silver atoms at the tips will interact with BSA, resulting in their separation from the original NPs, leading to morphological change from silver nanotriangles to nanodisks. TEM images clearly showed the morphological transition of the three morphological types of AgNPs after interaction with BSA, consistent with the results of UV-Vis spectroscopy and DLS results.

### 3.5. Protein-Mediated AgNP Aggregation

UV-Vis is one of the most commonly used methods to characterize the optical properties of AgNPs. The size, shape, and dispersion of AgNPs can be observed by the characteristic absorption peaks in UV-Vis [38]. The spectral variations of the effect of different morphologies of AgNPs on BSA are shown in Appendix A. It is found that the plasmon shift in AgNPs is normally interpreted as providing a measure of the distance between the particles, either as isolated particles where that scale is large or between the particles in aggregates where the distance is typically comparable to the particle radius. The absorbance of silver nanospheres at 420 nm decreased with the increase of BSA concentration without significant wavelength change (Appendix A), indicating the interaction between the silver nanospheres and BSA. BSA gradually coated the surface of silver nanospheres to form a protein corona. In addition, the transverse absorption peak at 425 nm and longitudinal absorption peak at 520 nm of silver nanorods also decreased in intensity with BSA concentration at constant wavelength (Appendix A). A previous study concluded that the changes observed in the spectra at around 425 and 520 nm showed that BSA was adsorbed not only on the sides, but also on both ends of silver nanorods [39]. The gradual adsorption of BSA on the surface of AgNPs reduces the availability of exposed AgNPs, leading to a decrease in absorbance. Appendix A shows the UV spectra of silver nanotriangles at different concentrations of BSA. The mixed solution of silver nanotriangles with BSA shows two obvious absorption peaks at 332 and 553 nm. The absorption peaks of silver nanotriangles are very different in the presence and absence of BSA. It is noted that the peak of silver nanotriangles at 483 nm disappears. At the same time, the addition of BSA results in delined absorbances at 628 and 332 nm, and an obvious blue shift of the peak at 628 nm to 553 nm. Moreover, the color of silver nanotriangles changed from blue to purple. These observations suggest that BSA had a marked influence on the shape and size of silver nanotriangles. Silver nanotriangles contain three sharp vertices, or tips, that contribute significantly to their optical and electronic properties [40,41]. In addition, silver nanotriangles have a high surface energy, particularly at the tip where the silver atoms are extremely unstable, and the particles can be easily rounded. The smoothing at the tips may result in color change of the silver nanotriangle dispersion and shift or disappearance of absorption peaks. The spectral studies indicate unambiguous interaction between silver nanotriangles and BSA, which induces the morphological transition from a triangular to disk shape.

DLS has already been used to describe protein–NP interactions and to determine changes in the size and distribution of NPs [42]. Adsorption of protein molecules onto the surface of NPs tends to increase the size of the NPs. Other groups have demonstrated that DLS is useful to monitor the specific and nonspecific adsorption of proteins onto AgNPs [43,44,45,46]. Based on the changes in size and shape of NPs, we used DLS to analyze the aggregation changes of AgNPs induced by protein adsorption (Figure 6). The average size of silver nanospheres after interaction with BSA was 89.69 nm (Figure 6D). The particle size increased slightly from 85.66 nm in the absence of BSA to 89.69 nm in its presence. This change in size of silver nanospheres suggest the adsorption of BSA molecules onto their surface, forming a protein corona. In addition, silver nanorods also enlarge from 87.28 nm to 91.14 nm by forming a protein corona (Figure 6E). In contrast to silver nanospheres and silver nanorods, silver nanotriangles showed a small decrease in particle size in the presence of BSA (Figure 6F). Thus, BSA bound to AgNPs with different morphologies in different ways. The smooth of the tips of silver nanotriangles lead to reduced size in the presence of BSA, which is consistent with the UV-Vis result.

## 4. Conclusions

Stability of AgNPs in physiological environments is a crucial factor for their application in bionanomedicine and bionanotechnology. The present study examined the roles of AgNP morphology and protein properties in determining the characteristics of the resultant nanobioconjugates. The interactions between AgNPs of different morphologies and BSA were monitored by a series of spectroscopic methods, DLS and TEM. The results showed that the combination of BSA to AgNPs with different morphologies led to a wide range of changes in spectral characteristics. Fluorescence spectra showed that static quenching occurred with complex formation between BSA and AgNPs with different morphologies. The conformational changes of BSA induced by AgNPs were analyzed by FTIR and CD. The different morphologies of AgNPs had different effects on the secondary structure of BSA, silver nanospheres and silver nanorods induced greater changes in the protein secondary structure relative to silver nanotriangles. In addition, AgNPs with different morphologies showed different protein-mediated aggregation behaviors by incorporation of BSA. Upon interacting with BSA, the surface of silver nanospheres and silver nanorods formed a protein corona. Moreover, silver nanotriangles showed a morphological evolution to nanodisks in presence of BSA, which may have been due to the instability of silver atoms at the tips (so that the triangles were easily rounded), as confirmed by TEM. The results presented herein provide further insight into the possible biological reactions and risks of AgNPs in biological systems and furnish a valuable reference for safe biomedical applications of AgNPs.

## Figures and Tables

**Figure 1 materials-16-05821-f001:**
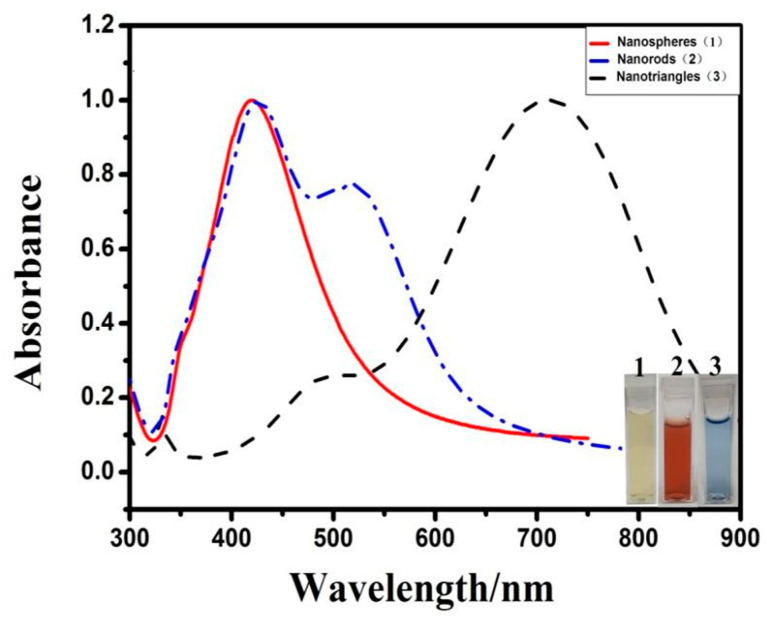
UV-Vis spectra of as-synthesized silver nanospheres, silver nanorods, and silver nanotriangles.

**Figure 2 materials-16-05821-f002:**
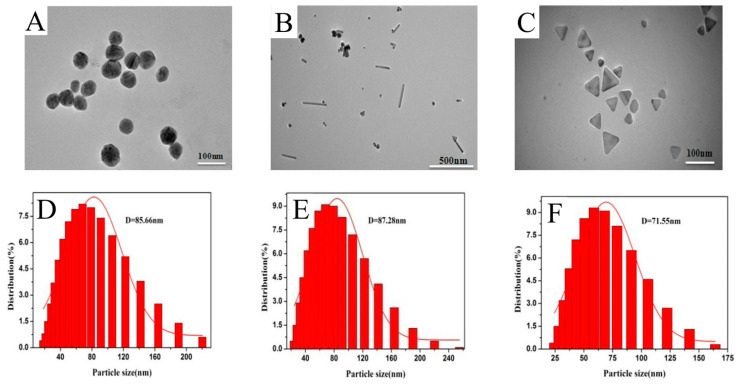
TEM of (**A**) silver nanospheres, (**B**) silver nanorods, and (**C**) silver nanotriangles. Histograms of the particle size distribution of (**D**) silver nanospheres, (**E**) silver nanorods, and (**F**) silver nanotriangles. The solid curve is the Gaussian fit to the histogram.

**Figure 3 materials-16-05821-f003:**
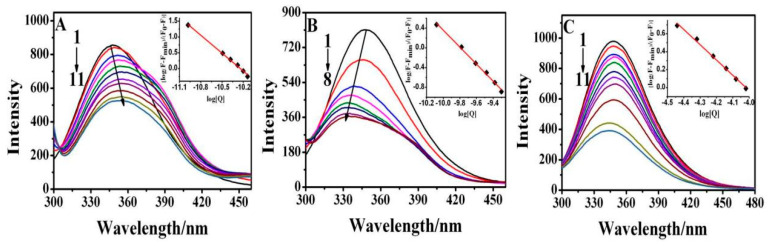
Fluorescence spectra of the interaction of BSA with (**A**) silver nanospheres: (1–11: 0, 0.01, 0.05, 0.10, 0.20, 0.30, 0.40, 0.60, 0.80, 0.90, 1.04) × 10^−10^ M, (**B**) silver nanorods: (1–8: 0, 1.0, 2.0, 3.0, 4.0, 5.0, 6.51, 7.45) × 10^−10^ M, and (**C**) silver nanotriangles: (1–11: 0, 0.02, 0.04, 0.08, 0.16, 0.32, 0.64, 0.80, 0.90, 1.09, 1.177) × 10^−4^ M.

**Figure 4 materials-16-05821-f004:**
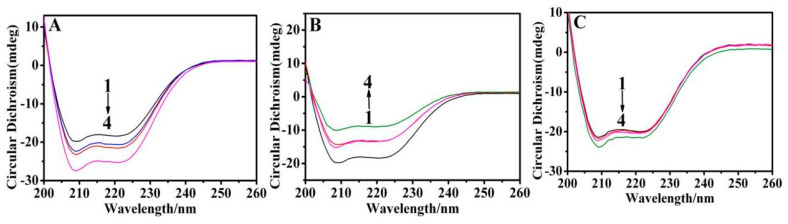
CD spectra of the interaction of BSA with (**A**) silver nanospheres (1–4: 0, 1 × 10^−12^ M, 2 × 10^−12^ M, 3.9 × 10^−12^ M), (**B**) silver nanorods (1–4: 0, 0.33 × 10^−10^ M, 0.615 × 10^−10^ M, 1.104 × 10^−10^ M), and (**C**) silver nanotriangles (1–4: 0, 0.336 × 10^−6^ M, 0.811 × 10^−6^ M, 1.801 × 10^−6^ M).

**Figure 5 materials-16-05821-f005:**
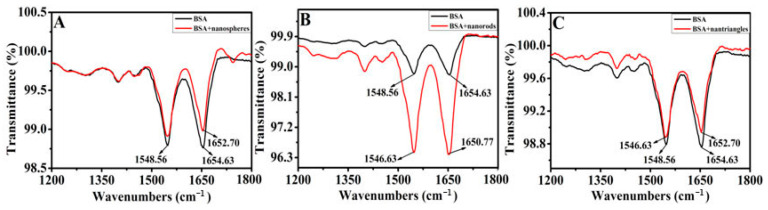
FTIR of the interaction of BSA with (**A**) silver nanospheres (0.052 nM), (**B**) silver nanorods (0.184 nM), and (**C**) silver nanotriangles (0.24 μM) in aqueous solution.

**Figure 6 materials-16-05821-f006:**
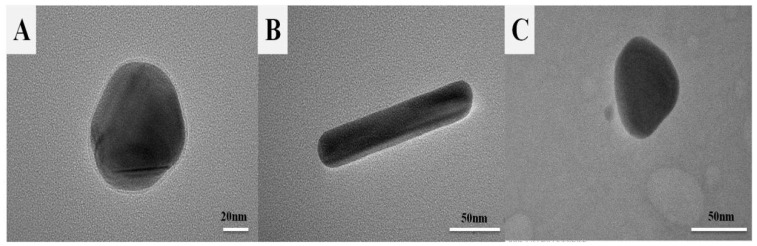
Representative TEM images of the interactions of BSA with (**A**) silver nanospheres, (**B**) silver nanorods, and (**C**) silver nanotriangles. Histograms of the particle size distribution in the interaction of BSA with (**D**) silver nanospheres, (**E**) silver nanorods, and (**F**) silver nanotriangles. The solid curve is a Gaussian fit to the histogram.

**Table 1 materials-16-05821-t001:** Hill coefficients and dissociation constants of the interaction between BSA and AgNPs with different morphologies.

System	*m*	*K_D_* (L·mol^−1^)	*R* ^a^
BSA–Nanospheres	1.89 ± 0.06	(5.5 ± 0.59) × 10^−11^	0.9955
BSA–Nanorods	1.72 ± 0.05	(1.55 ± 0.51) × 10^−10^	0.9953
BSA–Nanotriangles	1.70 ± 0.06	(9.57 ± 0.24) × 10^−5^	0.9940

*R*^a^: correlation coefficient.

## Data Availability

The data presented in this study are available on request from the corresponding author.

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
