# Peer review of "The Morphology Dependent Interaction between Silver Nanoparticles and Bovine Serum Albumin"

_materials, 2023, doi:10.3390/ma16175821_

Round 1
Reviewer 1 Report
The manuscript by Xianjun Fu et al. is devoted to the study of the effects of silver nanoparticle morphology on its interaction with bovine serum albumin. The authors should improve the manuscript to clarify some issues:
1. English should be improved. There are a lot of spelling mistakes in the text such as “Spectras” (Fig S1), "Aanospheres" (Name 2.2.1.) etc.
2. Figs 3 and S1 show UV–visible fluorescence and absorption spectra of interaction of BSA with silver NPs with different morphology for different BSA and NPs concentration ranges. The results presented is not correct to compare. Extended explanations are required.
3. Only one parameter affecting the interaction of Ag NPs and BSA that is the particle morphology is considered in the work. At the same time, the Ag particles were synthesized by various techniques, which could significantly affect the surface properties of nanoparticles, their zeta potential, sorption characteristics, and reactivity. Therefore, the findings made in the manuscript may be fallacies. The surface properties of nanoparticles should be considered (compared) in the article.
Author Response
1#
The manuscript by Xianjun Fu et al. is devoted to the study of the effects of silver nanoparticle morphology on its interaction with bovine serum albumin. The authors should improve the manuscript to clarify some issues:
- English should be improved. There are a lot of spelling mistakes in the text such as “Spectras” (Fig S1), "Aanospheres" (Name 2.2.1.) etc.
Response: We have carefully checked the manuscript throughout and invited a good English speaker to polish the language. Now, we have corrected any of the errors and avoided any of the possible mistakes.
- Figs 3 and S1 showfluorescence and UV–visible absorption spectra of interaction of BSA with silver NPs with different morphology for different BSA and NPs concentration ranges. The results presented is not correct to compare. Extended explanations are required.
Response: According to the reviewer’s suggestions, we have checked Fig 3 and Fig S1, and added the related explanations in the revised manuscript.
- Only one parameter affecting the interaction of Ag NPs and BSA that is the particle morphology is considered in the work. At the same time, the Ag particles were synthesized by various techniques, which could significantly affect the surface properties of nanoparticles, their zeta potential, sorption characteristics, and reactivity. Therefore, the findings made in the manuscript may be fallacies. The surface properties of nanoparticles should be considered (compared) in the article.
Response: According to the reviewer’s helpful suggestions, we have added the fluorescence spectroscopy experiments to estimate the effects of the different ligands onto AgNPs surface on the interaction between NPs and BSA. The surface ligands for spherical, rod-shaped and triangular AgNPs are sodium citrate, CTAB and PVP, respectively. As shown in Fig. R1, when added only the three surface ligands in protein solution, it is seen that the fluorescence of BSA was slightly quenched (curves 2-4). However, with the addition of spherical, rod-shaped and triangular AgNPs, the fluorescence of BSA decreased greatly (curves 5-7). Therefore, these results demonstrate that, compared to AgNPs themselves, the effects of different surface ligands of AgNPs on the interaction with proteins can be neglected in this case. These contents have been added in the revised manuscript (3.4 Morphological Analysis of three BSA–AgNPs Systems).
Fig. R1. The effects of surface ligands of AgNPs on the fluorescence spectroscopy of BSA. From 1-7: they represent BSA, 8 μM), sodium citrate (1 mM), CTAB (1 mM), PVP (1 mM), spherical AgNPs (7×10-10 M), rod-shaped AgNPs (1.5×10-9 M) and triangular AgNPs (5×10-4 M).

Reviewer 2 Report
At the introduction the authors’ said BSA is formed by 582 amino 73 acid residues with a molecular weight of 69,000, two tryptophan residues at positions 134 74 and 212. Authors should check other references and cite them.
In line 107, if the method employed to synthesis is very describe please put the relevant reference.
It is not understood under what conditions the BSA is placed with the nanoparticles to evaluate the interactions, how long if the supernatant is removed.
In the results, figures 3 and 4 cannot be understood by themselves, the image is not of good quality and there are no legends that indicate what each curve represents in the plot.
Can the author add the reference: in line 267 “Interaction of human serum albumin with silver nanoparticles functionalized with polyvinylthiol, Journal of Molecular Liquids 204 (2015)”.
In figure 2, from TEM it is not possible to easily read the scale of the size bar under which the NPs are evaluated, in the case of cylinders other circular shapes can be observed, under the synthesis conditions used, what percentage of nanoparticles They have the expected cylindrical shape. That percentage can be calculated.
Author Response
2#
At the introduction the authors’ said BSA is formed by 582 amino acid residues with a molecular weight of 69,000, two tryptophan residues at positions 134 and 212. Authors should check other references and cite them.
Response: We have checked the related references and cited them in the revised manuscript.
In line 107, if the method employed to synthesis is very describe please put the relevant reference.
Response: For the synthesis of AgNPs with different morphologies, We have referred to the related references, and put them in the experimental section. Moreover, we have appropriately added the corresponding experimental details in order to reach our experiment aims. These are shown in the section of 2.2. Synthesis of AgNPs with Different Morphologies.
It is not understood under what conditions the BSA is placed with the nanoparticles to evaluate the interactions, how long if the supernatant is removed.
Response: In phosphate-buffered saline (PBS, 0.01 M, pH 7.4), the emission spectra of BSA (2 μM) were measured in the presence or absence of AgNPs with different morphologies, at 298 K in the wavelength range of 300–550 nm. According to the fluorescence data, the Stern-Volmer equation is adopted to analyze the quenching mode, and the synergistic and binding constants of the interaction between BSA and AgNPs with different morphologies were evaluated by the Hill equation. The supernatant is removed in the process of AgNPs synthesis, however in the interaction of AgNPs with BSA, the mixed solutions were measured with a spectrofluorometer without centrifugal separation.
In the results, figures 3 and 4 cannot be understood by themselves, the image is not of good quality and there are no legends that indicate what each curve represents in the plot.
Response: In figures 3 and 4, for each curve, we have added the concentrations of AgNPs with different morphologies in the revised manuscript. Now, we believe these figures can be easily understood for the readers. Thank you for your helpful remind.
Can the author add the reference: in line 267 “Interaction of human serum albumin with silver nanoparticles functionalized with polyvinylthiol, Journal of Molecular Liquids 204 (2015)”.
Response: We have added this reference in the section of 3.2 Fluorescence Spectroscopy, furthermore we have cited it in the Introduction part in the revised manuscript.
In figure 2, from TEM it is not possible to easily read the scale of the size bar under which the NPs are evaluated, in the case of cylinders other circular shapes can be observed, under the synthesis conditions used, what percentage of nanoparticles They have the expected cylindrical shape. That percentage can be calculated.
Response: In figure 2, A, B and C represent the three morphologies, silver nanospheres, silver nanorods and silver nanotriangles, it is seen that AgNPs with different morphologies were well dispered, and have uniform size in certain degree. By using TEM images, the average dimensions of the AuNPs were determined by counting 80 NPs for each sample with Nanomeasurer software. Moreover, the particle sizes of AgNPs with three morphologies are approximately in accordance with ones measured by DLS. The added explanations are shown in the revised manuscript.

Reviewer 3 Report
Fu et al. examined the impact of AgNP morphology on the interaction with BSA protein. A variety of spectroscopic methods, DLS and TEM were used to analyze the interactions between silver nanospheres, silver nanorods, and silver nanotriangles with BSA. Results indicated that BSA combined with AgNPs of different morphologies produced a wide range of spectral features. The paper is well and can be published in materials after minor revision.
1-Could you please report the thermal stability (TGA) of the different AgNP morphologies
2- The mechanical behavior of different morphologies is important to include
3- The reviewer suggests reporting the biocompatibility of the designed materials
Author Response
3#
Fu et al. examined the impact of AgNP morphology on the interaction with BSA protein. A variety of spectroscopic methods, DLS and TEM were used to analyze the interactions between silver nanospheres, silver nanorods, and silver nanotriangles with BSA. Results indicated that BSA combined with AgNPs of different morphologies produced a wide range of spectral features. The paper is well and can be published in materials after minor revision.
- Could you please report the thermal stability (TGA) of the different AgNP morphologies.
Response: Thank you for your concerning this issue. After the preparation of AgNPs, we stored the sealed stock solution at 4 ℃ in a refrigerator. In order to evaluate the stability of the different AgNP morphologies, we measured UV-Vis spectra of AgNPs at regular interval days. The following figure (Figure R1) is shown the UV absorption of Ag nanosphere with the different storage days. We can see that the absorption of Ag nanosphere is almost constant with the same sample of 10 days storage, while the absorption seems to have a greater alteration after 15 days storage, indicating that the properties of Ag nanosphere are stable within 10 days. Ag nanorods and Ag nanotriangles have the similar trends. Based on these observations, we tried our best to carry out the related experiments with 10 days to avoid the properties changes of AgNPs. As for some experimental process, such as CD and FT-IR, which can not complete with 10 days, we re-synthesized AgNPs with the same method to keep the reliability of our experiments. Now, we have added the related information in 2.2. Synthesis of AgNPs with Different Morphologies section, which is shown in the revised manuscript.
Figure. R1 Absorbance of Ag nanosphere with the different storage time.
- The mechanical behavior of different morphologies is important to include
Response: The present work was mainly performed to examine the role of AgNP morphology, the change of secondary structure for BSA, in determining the characteristics of the resultant nanobioconjugates. Here, silver nanospheres, silver nanorods, and silver nanotriangles were synthesized by different methods, and the interactions between AgNPs with different morphologies and BSA was obtained by a number of spectroscopic methods, DLS, and TEM. The mechanical behavior of different morphologies of AgNPs is important and we will carry out the related investigations in the future work. Thank you!
3- The reviewer suggests reporting the biocompatibility of the designed materials
Response: Herein, we selected BSA as model protein, and analyzed the effects of different AgNPs on adsorbed proteins, the changes in protein structure, AgNPs morphology upon adsorption, and the resulting stability of NP–protein conjugates. It has not directly referred to the biological application and biological response, therefore the biocompatibility of AgNPs with the different morphologies was not emphasized in this work. Fortunately, it is know that both AgNPs and AdNPs have possessed the good biocompatibility, facilitating their applications in biological and medical fields (Refs 1-8, 12, 23, 25).

Round 2
Reviewer 1 Report
1. The section numbering in the paper should be checked and corrected.
2. The quality (resolution) of all (!!!) figures in the final document should be improved.
Author Response
- The section numbering in the paper should be checked and corrected.
Response: We have checked the section numbering carefully and corrected them.
- The quality (resolution) of all (!!!) figures in the final document should beimproved.
Response: We have remade all of the figures in the final manuscript, now the quality of all figures are greatly improved and suitable for the journal publication.
In addition, for Quality of English Language suggested by the reviewer, we have invited a good English speaker who studied abroad in the United States for many years to polish the English Language and sentence structures . Now, we believe that the English Language of the final manuscript has been suitable for the journal requirement. All the revisions are shown in red fonts in the revised manuscript.
